# Virtual Consultation in Dermatology: Access Inequalities According to Socioeconomic Characteristics and the Place of Residence

**DOI:** 10.3390/healthcare12060659

**Published:** 2024-03-14

**Authors:** Almudena Marco-Ibáñez, Carlos Aibar-Remón, Adriana Gamba-Cabezas, Lina Maldonado, Isabel Aguilar-Palacio

**Affiliations:** 1Primary Health Physician, Aragon Health Service, 50009 Zaragoza, Spain; amarcoi@salud.aragon.es; 2Health Services Research Group (GRISSA), Aragon Health Research Institute, 50009 Zaragoza, Spain; caibar@unizar.es (C.A.-R.); lagamba@iisaragon.es (A.G.-C.); lmguaje@unizar.es (L.M.); 3Preventive Medicine and Public Health, University of Zaragoza, 50009 Zaragoza, Spain; 4Department of Applied Economy, University of Zaragoza, 50009 Zaragoza, Spain; 5Network for Research on Chronicity, Primary Care, and Health Promotion (RICAPPS), ISCIII, 28029 Madrid, Spain

**Keywords:** teledermatology, virtual consultation, inequalities, multilevel model

## Abstract

Virtual consultation has been implemented as a tool to improve the cooperation and coordination between primary care and other specialties. However, in its use in dermatology, inequities have been described. The aim of this study was to identify individual and geographical factors affecting the likelihood of accessing this resource. We conducted a cross-sectional study. The study population was individuals living in Aragón, a region in the north-east of Spain, who were users of the Aragon Health Service in 2021. To explore the differences in individual and area characteristics, between patients with virtual and non-virtual dermatology consultation, we performed bivariate analyses. To analyse the use of virtual consultation in dermatology, a multilevel methodology stratified by sex was developed. We analysed 39,174 dermatology visits, with 16,910 being virtual (43.17%). The rates of virtual consultation were higher in urban areas and the most affluent areas, for older persons, chronic complex patients and people with more advantageous socioeconomic positions. The best multilevel model conducted showed inequalities by socioeconomic position and area of residence. There are individual and area inequalities in the use of teledermatology. As this tool should improve equity of access, teledermatology interventions must address and adapt to the needs of the local patient population.

## 1. Introduction

In recent years, in order to improve the cooperation and coordination between primary care and other specialties, as well as the continuity of care, virtual consultation has been implemented around the world as a telemedicine tool. This is an asynchronous, bidirectional, online and on-demand communication tool. It is usually based on the issuing of a telematic collaboration request from primary care, which is subsequently attended to by a doctor from the specialty to which the request has been addressed [1]. Its use likely avoids unnecessary patient travel [2] and reduces waiting times, facilitating the care of dispersed and remote populations and users with impaired functional health [3]. The initial decision to request a virtual or non-virtual consultation is at the discretion of the primary care physician, based on the patient’s clinic and physical examination. Nonetheless, patient preferences are also taken into account. 

In the case of a virtual consultation in dermatology, also known as teledermatology, it consists of sending an image of the skin lesion under study together with its clinical description and the patient’s history for joint assessment [4]. In Aragon, a virtual consultation to dermatology is an asynchronous, bidirectional, written online, on-demand communication tool, which is based on the issuing of a telematic collaboration request from the primary care physician, attaching an image of the patient’s lesion and its description, and which is responded to through specialised care from the dermatology consultation. Poor quality of the attached image can lead to diagnostic errors.

Teledermatology allows for diagnostic affiliation, triage of lesions suspected of malignancy and the earlier initiation of appropriate treatment [5]. The COVID-19 pandemic prompted even greater expansion and implementation of teledermatology [6], which also facilitates education among providers at different levels of training [7]. In the case of rural areas, the addition of teledermoscopy use by patients in remote self-skin examinations (SSEs) may enhance the triage of patient-selected lesions of concern by providing digital dermoscopy images to dermatology providers [8]. As for urban areas, teledermatology can efficiently provide care to outpatient populations in these settings and can expand access to care in regions with higher dermatologist density [6]. However, not having an adequate internet connection, dermatoscopes or cameras to take quality clinical images is a barrier to the use of virtual consultation in dermatology. In Aragon, the high geographical dispersion, the progressive population aging and the limited specialised healthcare resources, centralised in urban areas, pose a major challenge for healthcare services. In this context, the use of in-person dermatology is reserved for the most severe, surgical or complex diseases.

However, the universal implementation of these digital health services may present inequalities in health care in some patient groups [9], especially those with less social support and less technological knowledge. In the specific case of dermatology and virtual consultations, inequities in their use related to socioeconomic determinants have been described. Initially, it could be thought that patients from geographically isolated areas [10] and vulnerable groups could benefit the most from this type of resource. However, living in rural areas [8], not having the necessary technological means [11] or belonging to an ethnic minority [12] have been associated with a lower use of virtual consultations, especially if one is not fluent in the language of the country of residence [13]. In addition, it is known that individuals of lower socioeconomic status and education levels, the elderly and uninsured patients have poorer melanoma and non-melanoma skin cancer outcomes, and atopic dermatitis is more prevalent in minorities [14]. This, combined with reduced use of virtual consultations, could lead to increased inequalities in care for certain groups.

Assessing access to teledermatology, and its possible inequities at the individual and territorial levels, is key to implementing quality health services. For this reason, the aim of this study was to analyse the use of virtual consultation in dermatology in Aragon, a Spanish region in south-eastern Europe, to identify individual and geographical factors affecting the likelihood of accessing this resource.

## 2. Materials and Methods

### 2.1. Design, Study Population and Data Sources

We conducted a cross-sectional study. The study population encompassed individuals living in Aragón and, thus, users of the Aragon Health Service in 2021. Aragon is located in the north-east of Spain and has almost 1.5 million inhabitants; its capital is Zaragoza, where most of the specialized medical care resources are centralized. The Spanish health system is mainly tax financed and is based on universality, free health care, equity and fairness in financing [15].

Data about all inter-consultations carried out in Aragon in this year were proportioned by the Aragon Health Department. Data from the Users’ Database (BDU) and the Adjusted Morbidity Groups (GMA) were added in order to obtain sociodemographic and clinical information of users.

Data were proportioned fully anonymized by the Department of Health of the Aragon Health Service. Personal consent was not required. Approval was obtained from the Research Ethics Committee of the Community of Aragón (CEICA), PI20-334.

### 2.2. Variables of the Study

In order to conduct this study, we used information about inter-consultations and users. Regarding inter-consultations, we obtained the number of virtual and non-virtual consultations in dermatology by Basic Healthcare Area (BHA) in Aragón in 2021.

Regarding sociodemographic characteristics, we consider the following individual characteristics: sex; age, both as a quantitative variable and categorized in 5 groups (≤15 years old; 16 to 44; 45 to 64; 65 to 79; ≥80); socioeconomic status, which was determined for each individual based on a combination of their level of pharmaceutical co-payment and their type of economic activity, resulting in five mutually exclusive categories; chronic complex patient; GMA weight categorized in 3 groups (people with GMA weight; >99th percentile (high-morbidity patients); GMA weight from 97th to 99th percentile; GMA weight <97th percentile).

We also considered the following characteristics of the place of residence: geographical dispersion of primary care teams. This index considers both the number of primary care professionals and the average distance of the population centres from the head municipality. The result is presented in 4 categories, with G1 being those with a single population nucleus and G4 those with greater dispersion; rural or urban BHA, according to the Aragon Government [16], with urban areas being those that concentrate at least 80% of the BHA population in their municipalities and rural areas as those that do not meet this criterion; depopulation level of the BHA was assigned based on the criteria of the Spanish Ministry for the Ecological Transition and the Demographic Challenge [17], who defines depopulated municipalities as those with fewer than 5000 inhabitants; BHA deprivation index categorized into 4 quartiles (least (Q1) to most (Q4) deprived). This deprivation index combines information of four indicators from the Population and Housing Census: percentage of unemployment, percentage of temporary workers, percentage of people between 16 and 64 years with low educational level and percentage of immigrants [18].

### 2.3. Analysis

First, we selected those BHAs where both virtual and non-virtual consultations coexisted in 2021. This criterion was met in 58 of the 123 existing zones. Then, we obtained the use rates (%) of virtual and non-virtual consultations by BHAs in Aragón. Mean and standard deviation (SD) were used to describe continuous variables, and frequencies and percentages were used to describe categorical variables.

To explore differences in individual and area characteristics between those patients with a virtual and non-virtual dermatology consultation, we performed bivariate analyses. Statistical differences were assessed using chi-square (categorical variables) and Mann–Whitney tests (continuous variable). Finally, to analyse the use of virtual consultation in dermatology, a multilevel methodology stratified by sex was developed, considering individual characteristics and characteristics of the BHA of residence. In this case, we have a two-level model, with a cross-classification structure; at level 1 are the patients, and at level 2, we have the deprivation index and the depopulation level of the BHA in which a patient resides. Given the characteristics of this study, there is an intraclass correlation, which means that there are observations that are more similar to others in the same group than to those in other groups. Variance partition coefficients can be calculated to see how much of the response variance belongs to each level. To evaluate statistical significance, a *p*-value smaller than 0.05 was used.

Individuals could simultaneously belong to more than one group at a given hierarchical level. Thus, at the same time, an individual belongs to a BHA with a certain deprivation index and to a BHA with a certain level of depopulation. This leads to a cross-classification structure. In this case, we classify the virtual consultation cases by their deprivation index (quartiles) and level of depopulation, so both are considered random. Cross-random effects are used when each category of one factor co-exists with each category of the other factor (there is at least one category observation for both factors). The model we have is as follows:$$VC_{i\left(sj\right)}=\log\frac{\pi_{sj}}{1-\pi_{sj}}=\beta_{0}+{\left(X\beta \right)}_{i\left(sj\right)}+u_{s}+u_{j}+e_{i\left(sj\right)}$$
with $\pi_{sj}=P\left(y_{sj}=1\right)$ being the probability that a patient has a virtual consultation, when an individual i belongs to a BHA with a level of depopulation s(s = 1 (no depopulated municipality), 2 (some depopulated municipality), 3 (all depopulated municipalities)) and with a deprivation index j (j = 1,…4 quartiles-). In this model, X is the set of explanatory variables. Individual sociodemographic characteristics (age, socioeconomic level) and GMA and PCC (complex chronic patient) were considered as explanatory variables. The parameter β represents the fixed effects. This model has three assumptions: first, the random effects $u_{s}$ and $u_{j}$ are normally distributed with mean 0 and variance $\sigma_{u}^{2}$; second, the error component $e_{i\left(sj\right)}$ is also normally distributed with mean 0 and variance $\sigma_{e}^{2}$; third, the random effects $u_{s}$ and $u_{j}$ and the error component $e_{i\left(sj\right)}$ are independent, and $e_{i\left(sj\right)}$ are all independent of each other. Interactions between variables were systematically investigated, and collinearity was demonstrated. Finally, the likelihood ratio test (LR test) was used to evaluate the final model. The significance of the fixed effects was also evaluated with the Wald Test. All analyses were performed using R statistical software (the R Foundation for Statistical Computing, Vienna, Austria). Data were analysed using mixed-effects linear regression based on the lme4 [19] package in the statistical package R version 4.3.1.

## 3. Results

Data from virtual consultation in dermatology in Aragón in 2021 were analysed. In particular, 58 from 123 BHAs in Aragón had totally working virtual and non-virtual consultations in dermatology. We analysed a total of 39,174 visits, with 16,910 being virtual (43.17%). In Table 1, a description of the total analysed sample can be found.

In Figure 1, the use rates of virtual and non-virtual consultation (%) by BHA can be observed. We found high geographic variability in virtual and non-virtual consultations in dermatology. The rates of virtual consultation were higher in urban than in rural areas, while non-virtual consultation presented higher rates in rural areas. In the city of Zaragoza, large variability between BHAs was observed, with no pattern identified. Use rates were calculated only in those regions where both virtual and non-virtual consultations coexisted in 2021.

We analysed individual and area characteristics in those patients with a virtual or non-virtual consultation in dermatology. As can be observed in Table 1, regarding individual characteristics, we found statistically significant differences by age (*p* < 0.001), with the people who use virtual consultation being slightly older (50.8 years old vs. 48.1). We also observed statistically significant differences by socioeconomic position (*p* < 0.001). People with more advantageous socioeconomic positions used virtual consultation more frequently than less affluent people. The frequency of virtual consultation was higher in PCC (*p* < 0.001), but the number of these patients was very low, and no differences were observed by morbidity weight (*p* = 0.2274). Regarding area variables, the most affluent areas had a higher frequency of virtual consultation, while the most deprived presented a higher frequency of non-virtual consultation, with these differences being statistically significant (*p* < 0.001). Urban areas had a higher frequency of virtual consultation than rural areas (*p* < 0.001), as well as those areas with no depopulated municipalities (*p* < 0.001) (Table 2).

In Table 3 and Table 4, bivariate analyses stratified by sex can be observed.

We tested different multilevel models, combining variables of the area of residence in order to obtain the model with the highest explanatory capacity, both for the total population and stratified by sex. The best model was the one that combined the deprivation index and depopulation level of the BHA. The results of this model can be found in Table 5. In the adjusted models, we observed that the probability of obtaining a virtual consultation increased with age, with the group aged from 65 to 79 years old being the one with the highest probability of using virtual consultation (odds ratios (OR) 1.62; 95% confidence interval (95%CI) 1.47–1.79). There were also statistically significant differences by socioeconomic status. So, people employed earning more than EUR 18K per year presented the highest probability of virtual consultation. PCC had a higher risk of having a virtual consultation than non-complex patients (OR: 1.60; 95%CI 1.18–2.17). These differences were statistically significant for all the population analysed and for women, but no differences were observed for men (*p*: 0.361). Further, 11 patients were excluded from our analyses because their socioeconomic information was not available.

The influence of area variables can be observed in Figure 2. So, less deprived areas showed a higher probability of virtual consultation in dermatology than more deprived areas. According to the depopulation level, no unpopulated areas presented the highest risk of virtual consultation.

Random effects by men and women did not show differences. These results are available in Figure 3 and Figure 4.

## 4. Discussion

In recent years, teledermatology has advanced exponentially in acceptance and use in all health systems [20]. The results of this study show a high use of referrals for dermatology in 2021 in Aragon. A total of 39,174 visits were recorded in a single year, with 16,910 being virtual (43.17%). By comparison, Van der Heijden et al. [21] in the Netherlands, from March 2007 to September 2010, registered a greater use of teledermatology because, out of a total of 37,207 referrals, 26,596 (71.48%) were virtual. These differences could be due to the fact that general practitioners used teleconsultation to prevent a referral, not for a second opinion or to make a direct referral to dermatology.

When we analysed the sociodemographic characteristics of patients referred in Aragon during the study period, we observed differences, at the individual level and by area of residence, in the probability of receiving a virtual consultation compared to non-virtual. People who are older, have more comorbidities, higher socioeconomic status and live in urban areas are more likely to receive a virtual consultation. On the contrary, patients who live in more deprived areas are less likely to benefit from a virtual consultation.

Higher rates of telehealth among women and older persons have been observed. During the COVID-19 pandemic, higher rates of telehealth among older persons may have been due to their greater fear of contracting COVID-19 and the subsequent desire to quarantine and maintain social distance [22] but also to the doctors’ attempt to prevent them from unnecessary travel [23]. Since the end of the COVID-19 pandemic, we could explain this fact by the objective of doctors to leave the use of face-to-face dermatology appointments for more severe, surgical or complex diseases [24]. On the other hand, elderly patients seen in primary care are increasingly older, polypharmacological and with comorbidities, but they also present greater functional and mental impairment [25]. Despite this, more than 90% of their health problems are resolved in the community environment [26]. In this context, fluid interdisciplinary communication is key to improving care coordination and patient maintenance in their usual environment, and virtual interconsultation is a useful tool to achieve this [23], which could explain its greater probability of use in these patients.

Among patients of the same age, living in the same area of residence and with similar comorbidities, the highest likelihood of using virtual interconsultation was observed in people with high socioeconomic status. These differences may be due to their higher level of education, which helps them to detect clinical conditions earlier, and their easier access to the necessary equipment [6], which improves their health outcomes [14].

On the contrary, the existence of a language barrier between a doctor and patient can be a bias [10], as it hinders the exchange of clinical information and the understanding of patient preferences. At the same time, a lower use of virtual consultation by some professionals can be explained by their concern about the increase in administrative work [27] derived from their management or their limited computer experience [28]. These results are consistent with studies such as the one by Pierce and Stevermer [22], who also recorded that telehealth visits were used more often by self-pay status but less often by those of non-white race and those from rural postal codes, because rural residents have fewer healthcare services, fewer trained physicians and worse broadband coverage. Nonetheless, these results strongly depend on the context and health coverage available. For example, Armstrong et al. [29] noted that, in California, over 75% of patients were at or below the 200% federal poverty level and usually lived in rural regions without direct or easy access to dermatologists, since teledermatology avoids unnecessary travel and face-to-face visits and their cost.

Once the other characteristics of the individual have been controlled, the fact that urban areas have an increased likelihood of virtual consultations could be explained by the increased availability of technological means to attach images to referrals and also by the increased density of dermatologists to assess them [6]. However, this is contrary to the initial objective of this tool, which is to facilitate care for dispersed and remote populations [30,31], reducing access inequalities [32] to healthcare due to the heterogeneous and centralized distribution of limited resources, both physical and human. As a consequence, patients who live in rural [8] and more deprived [10] areas are at higher risk of receiving lower quality, less effective, equitable and timely care, which can negatively influence their health outcomes. This is especially relevant in geographical areas such as the one in which this study was carried out, with high geographical dispersion and low population density.

This study is not without limitations. It is possible that there are repeated consultations with the same patient, with the possible overestimation of some results. On the other hand, this study focused on a region in south-eastern Europe, characterized by a marked geographical dispersion, not optimal communications and a densely populated capital, which could limit the generalization of the results to other territories with different characteristics. However, the high number of consultations and variables analysed from the target population and the fact that both urban and rural areas with different demographic characteristics could be assessed support the representativeness of the study and the validity of the results to identify individual and geographical factors affecting the likelihood of accessing a virtual dermatology consultation.

Since the universal implementation of digital health services can lead to inequalities in health care in some groups of patients [9], it is necessary to ensure their equitable representation in the design process of these tools. The Multidimensional Readiness and Enablement Index for Health Technology (READHY), comprising the eHealth Literacy Questionnaire (eHLQ), Health Literacy Questionnaire (HLQ) and Health Education Impact Questionnaire (heiQ), can be used to assess patient skills, confidence and experience in using technology to manage their health [9]. Regarding professionals, having real-time access to interpretation services to overcome the language barrier [10,12], screening for skin lesions in patients with risk factors [12], increasing training in dermoscopy and digital image archiving and transmission [33] are useful strategies to improve the use of virtual interconsultation.

Despite the advantages of teledermatology, its use in different demographic areas, for patients with different socioeconomic characteristics, may promote inequalities. Strategies such as the use of standardised teledermatology consult templets and the provision of real-time computer support could improve the effectiveness of this tool [7] and increase the resolution of primary care teams [34].

## 5. Conclusions

The main objectives of teledermatology are to improve equity of access to this specialty and to allow for a reduction in diagnostic and therapeutic delays. However, patients who live in rural and more deprived areas, and those with a lower socioeconomic level, are less likely to benefit from this tool. In order to reduce disparities and to improve the effectiveness of this resource, teledermatology interventions should address and adapt to the needs and characteristics of the local patient population.

## Figures and Tables

**Figure 1 healthcare-12-00659-f001:**
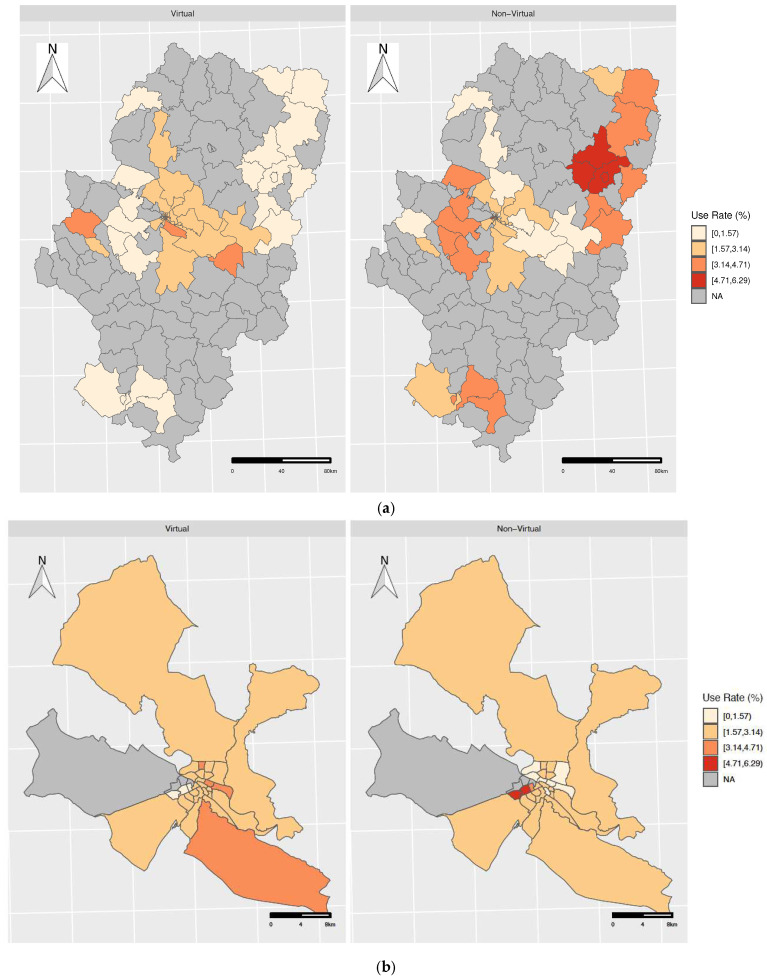
Geographic distribution of virtual and non-virtual consultation in (**a**) Aragón and (**b**) in the city of Zaragoza.

**Figure 2 healthcare-12-00659-f002:**
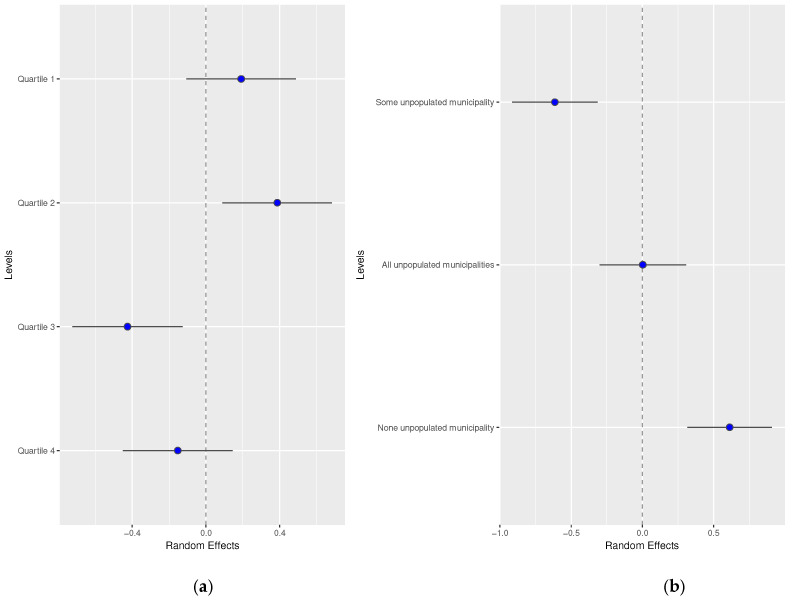
Random effects by (**a**) basic healthcare area deprivation and (**b**) by depopulation level of the basic healthcare area.

**Figure 3 healthcare-12-00659-f003:**
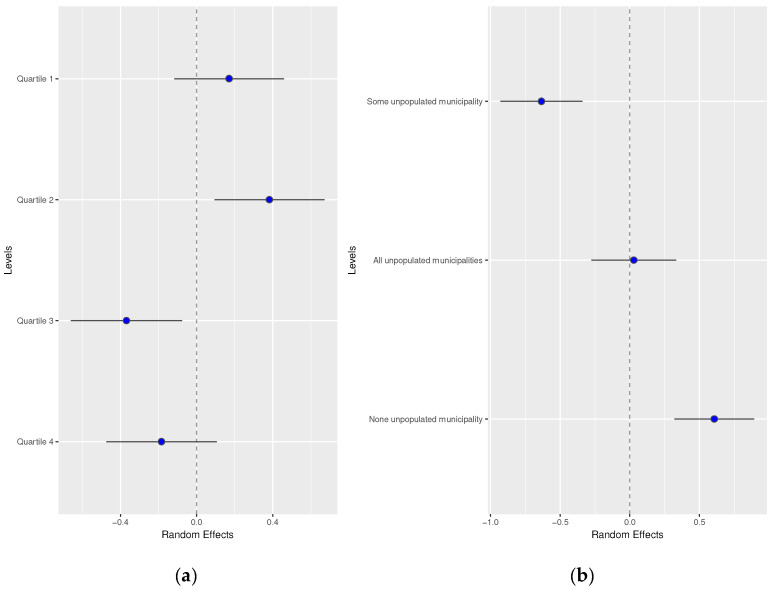
Random effects by (**a**) basic healthcare area deprivation and (**b**) by depopulation level of the basic healthcare area in men.

**Figure 4 healthcare-12-00659-f004:**
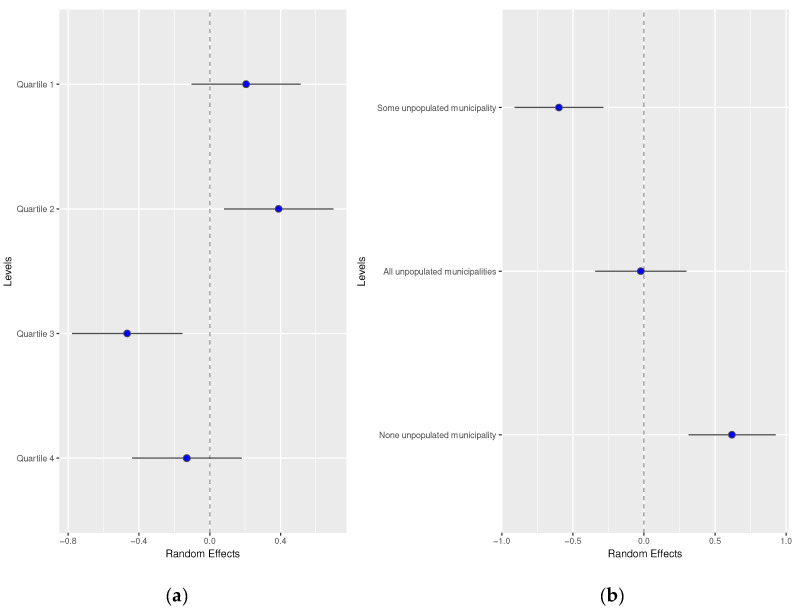
Random effects by (**a**) basic healthcare area deprivation and (**b**) by depopulation level of the basic healthcare area in women.

**Table 1 healthcare-12-00659-t001:** Description of the total sample analysed.

		N (%)/Mean (SD)
Individual variables	Sex	
	Men	16,848 (43.0)
	Women	22,326 (57.0)
	Mean age (SD)	49.3 (23.8)
	Groups of age (years old)	
	≤15	4031 (10.3)
	16–44	12,250 (31.3)
	45–64	10,543 (26.9)
	65–79	8333 (21.3)
	≥80	4017 (10.3)
	Socioeconomic level	
	Employed < 18K EUR per year	12,593 (32.2)
	Employed ≥ 18K EUR per year	8546 (21.8)
	Pensioner < 18K EUR per year	10,690 (27.3)
	Pensioner ≥ 18K EUR per year	5359 (13.7)
	Other	1976 (5.0)
	GMA	
	A (percentile >99%)	894 (2.3)
	B (percentile 97–99)	733 (1.9)
	C (rest)	37,547 (95.9)
	PCC	205 (0.5)
Area Variables	Geographical dispersion	
	1 (least dispersed)	25,713 (65.7)
	2	3499 (8.9)
	3	6737 (17.2)
	4 (most dispersed)	3224 (8.2)
	Zone of residence	
	Urban	32,755 (83.6)
	Rural	6419 (16.4)
	Depopulation level	
	No depopulated municipality	28,652 (73.2)
	Some depopulated municipality	7838 (20.0)
	All depopulated municipalities	2684 (6.9)
	Deprivation index	
	1 (least deprived)	12,565 (32.1)
	2	10,146 (25.9)
	3	7130 (18.2)
	4 (most deprived)	9333 (23.8)

N: number; % percentage; p: statistical signification; SD: standard deviation; GMA: Adjusted Morbidity Groups; PCC: complex chronic patient.

**Table 2 healthcare-12-00659-t002:** Description of virtual and non-virtual consultation in dermatology in Aragón in 2021. Bivariate analyses.

		Virtual	Non-Virtual	*p*
		N (%)/Mean (SD)	N (%)/Mean (SD)	
Individual variables	Sex			
	Men	7299 (43.16)	9549 (42.89)	0.5946
	Women	9611 (56.84)	12,715 (57.11)	
	Mean age (SD)	50.80 (23.30)	48.1 (24.20)	<0.001
	Groups of age (years old)			
	≤15	1401 (8.29)	2630 (11.81)	<0.001
	16–44	5182 (30.64)	7068 (31.75)	
	45–64	4652 (27.51)	5891 (26.46)	
	65–79	3867 (22.87)	4466 (20.06)	
	≥80	1808 (10.69)	2209 (9.92)	
	Socioeconomic level			
	Employed < 18K EUR per year	5122 (30.30)	7471 (33.57)	<0.001
	Employed ≥ 18K EUR per year	3793 (22.43)	4753 (21.36)	
	Pensioner < 18K EUR per year	4653 (27.52)	6037 (27.12)	
	Pensioner ≥ 18K EUR per year	2519 (14.90)	2840 (12.76)	
	Other	820 (4.85)	1156 (5.19)	
	GMA			
	A (percentile >99%)	411 (2.43)	483 (2.17)	0.2274
	B (percentile 97–99)	318 (1.88)	415 (1.86)	
	C (rest)	16,181 (95.69)	21,366 (95.97)	
	PCC	113 (0.67)	92 (0.41)	<0.001
Area variables	Geographical dispersion			
	1 (least dispersed)	12,555 (74.25)	13,158 (59.10)	<0.001
	2	1773 (10.48)	1726 (7.75)	
	3	1833 (10.84)	4904 (22.03)	
	4 (most dispersed)	749 (4.43)	2475 (11.12)	
	Zone of residence			
	Urban	15,078 (89.17)	17,677 (79.40)	<0.001
	Rural	1832 (10.83)	4587 (20.60)	
	Depopulation level			
	No depopulated municipality	14,306 (84.60)	14,346 (64.44)	<0.001
	Some depopulated municipality	1612 (9.53)	6226 (27.96)	
	All depopulated municipalities	992 (5.87)	1692 (7.60)	
	Deprivation index			
	1 (least deprived)	6022 (35.61)	6543 (29.39)	<0.001
	2	5278 (31.21)	4868 (21.86)	
	3	2041 (12.07)	5089 (22.86)	
	4 (most deprived)	3569 (21.11)	5764 (25.89)	

N: number; % percentage; *p*: statistical signification; SD: standard deviation; PCC: complex chronic patient; GMA: Adjusted Morbidity Groups.

**Table 3 healthcare-12-00659-t003:** Description of virtual and non-virtual consultation in dermatology in Aragón in 2021 in men. Bivariate analyses.

		Virtual	Non-Virtual	*p*
		N (%)/Mean (SD)	N (%)/Mean (SD)	
Individual variables	Mean Age (SD)	50.70 (23.60)	47.40 (24.60)	<0.001
	Groups of age (years old)			
	≤15	675 (9.25)	1247 (13.06)	<0.001
	16–44	2160 (29.59)	3028 (31.71)	
	45–64	1975 (27.06)	2413 (25.27)	
	65–79	1693 (23.19)	1923 (20.14)	
	≥80	796 (10.91)	938 (9.82)	
	Socioeconomic level			
	Employed < 18K EUR per year	1855 (25.42)	2857 (29.93)	<0.001
	Employed ≥ 18K EUR per year	2042 (27.98)	2510 (26.29)	
	Pensioner < 18K EUR per year	1734 (23.76)	2216 (23.21)	
	Pensioner ≥ 18K EUR per year	1388 (19.02)	1545 (16.18)	
	Other	278 (3.81)	418 (4.38)	
	GMA			
	A (percentile > 99%)	180 (2.47)	202 (2.12)	0.1563
	B (percentile 97–99)	147 (2.01)	169 (1.77)	
	C (rest)	6972 (95.52)	9178 (96.11)	
	PCC	51 (0.70)	48 (0.50)	0.1216
Area variables	Geographical dispersion			
	1 (least dispersed)	5361 (73.45)	5587 (58.51)	<0.001
	2	777 (10.65)	760 (7.96)	
	3	816 (11.18)	2139 (22.40)	
	4 (most dispersed)	345 (4.73)	1062 (11.12)	
	Zone of residence			
	Urban	6468 (88.61)	7481 (78.34)	<0.001
	Rural	831 (11.39)	2068 (21.66)	
	Depopulation level			
	No depopulated municipality	6131 (84.00)	6081 (63.68)	<0.001
	Some depopulated municipality	712 (9.75)	2720 (28.48)	
	All depopulated municipalities	456 (6.25)	748 (7.83)	
	Deprivation index			
	1 (least deprived)	2597 (35.58)	2819 (29.52)	<0.001
	2	2280 (31.24)	2084 (21.82)	
	3	931 (12.76)	2177 (22.80)	
	4 (most deprived)	1491 (20.43)	2469 (25.86)	

N: number; % percentage; *p*: statistical signification; SD: standard deviation; PCC: complex chronic patient; GMA: Adjusted Morbidity Groups.

**Table 4 healthcare-12-00659-t004:** Description of virtual and non-virtual consultation in dermatology in Aragón in 2021 in women. Bivariate analyses.

		Virtual	Non-Virtual	*p*
		N (%)/Mean (SD)	N (%)/Mean (SD)	
Individual variables	Mean Age (SD)	50.90 (23.00)	48.70 (23.80)	<0.001
	Groups of age (years old)			
	≤15	726 (7.55)	1383 (10.88)	<0.001
	16–44	3022 (31.44)	4040 (31.77)	
	45–64	2677 (27.85)	3478 (27.35)	
	65–79	2174 (22.62)	2543 (20.00)	
	≥80	1012 (10.53)	1271 (10.00)	
	Socioeconomic level			
	Employed <18K EUR per year	3267 (34.00)	4614 (36.30)	<0.001
	Employed ≥18K EUR per year	1751 (18.22)	2243 (17.65)	
	Pensioner <18K EUR per year	2919 (30.37)	3821 (30.06)	
	Pensioner ≥18K EUR per year	1131 (11.77)	1295 (10.19)	
	Other	542 (5.64)	738 (5.81)	
	GMA			
	A (percentile > 99%)	231 (2.40)	281 (2.21)	0.4486
	B (percentile 97–99)	171 (1.78)	246 (1.93)	
	C (rest)	9209 (95.82)	12188 (95.86)	
	PCC	62 (0.65)	44 (0.35)	0.0018
Area variables	Geographical dispersion			
	1 (least dispersed)	7194 (74.85)	7571 (59.54)	<0.001
	2	996 (10.36)	966 (7.60)	
	3	1017 (10.58)	2765 (21.75)	
	4 (most dispersed)	404 (4.20)	1413 (11.11)	
	Zone of residence			
	Urban	8610 (89.58)	10196 (80.19)	<0.001
	Rural	1001 (10.42)	2519 (19.81)	
	Depopulation level			
	No depopulated municipality	8175 (85.06)	8265 (65.00)	<0.001
	Some depopulated municipality	900 (9.36)	3506 (27.57)	
	All depopulated municipalities	536 (5.58)	944 (7.42)	
	Deprivation index			
	1 (least deprived)	3425 (35.64)	3724 (29.29)	<0.001
	2	2998 (31.19)	2784 (21.90)	
	3	1110 (11.55)	2912 (22.90)	
	4 (most deprived)	2078 (21.62)	3295 (25.91)	

N: number; % percentage; *p*: statistical signification; SD: standard deviation; PCC: complex chronic patient; GMA: Adjusted Morbidity Groups.

**Table 5 healthcare-12-00659-t005:** Probability of having a virtual consultation. Multilevel analyses stratified by sex.

	General Population		Men		Women	
	OR (95%CI)	*p*	OR (95%CI)	*p*	OR (95%CI)	*p*
Intercept	0.37 (0.19–0.74)	0.005 *	0.40 (0.19–0.82)	0.013 *	0.35 (0.17–0.72)	0.005 *
Groups of age (Ref: ≤15)						
16–44	1.32 (1.22–1.43)	<0.001 *	1.26 (1.12–1.41)	<0.001 *	1.37 (1.23–1.52)	<0.001 *
45–64	1.44 (1.33–1.56)	<0.001 *	1.46 (1.30–1.64)	<0.001 *	1.42 (1.28–1.59)	<0.001 *
65–79	1.62 (1.47–1.79)	<0.001 *	1.62 (1.39–1.88)	<0.001 *	1.62 (1.42–1.85)	<0.001 *
≥80	1.58 (1.42–1.77)	<0.001 *	1.58 (1.33–1.87)	<0.001 *	1.59 (1.37–1.84)	<0.001 *
Socioeconomic level (Ref: Employed ≥ 18K EUR)						
Employed < 18K EUR per year	0.89 (0.84–0.94)	<0.001 *	0.85 (0.78–0.93)	<0.001 *	0.91 (0.84–0.99)	0.032 *
Pensioner < 18K EUR per year	0.88 (0.81–0.95)	0.001 *	0.89 (0.79–1.00)	0.054	0.88 (0.79–0.98)	0.016 *
Pensioner ≥ 18K EUR per year	0.88 (0.80–0.96)	0.005 *	0.87 (0.76–1.00)	0.048 *	0.87 (0.77–0.99)	0.040 *
Other	0.87 (0.78–0.96)	0.007 *	0.81 (0.69–0.97)	0.018 *	0.91 (0.79–1.04)	0.152
GMA (Ref: Percentile > 99%)						
Percentile 97–99%	0.92 (0.74–1.13)	0.406	0.99 (0.73–1.36)	0.973	0.86 (0.65–1.13)	0.284
Rest	1.07 (0.92–1.24)	0.398	1.04 (0.83–1.31)	0.744	1.09 (0.89–1.33)	0.411
PCC (Ref: no)						
Yes	1.60 (1.18–2.17)	0.002 *	1.23 (0.79–1.91)	0.361	2.02 (1.32–3.08)	0.001 *
Random effects						
τ_00_	0.12_BHA deprivation_		0.11_BHA deprivation_		0.13_BHA deprivation_	
	0.27_Depopulation level_		0.28_Depopulation level_		0.27_Depopulation level_	
ICC	0.11		0.11		0.11	
Number of observations	39,163		16,842		22,321	
Marginal R^2^/Conditional R^2^	0.005/0.112		0.007/0.111		0.004/0.113	
Deviance	50,354.537		21,670.986		28,695.220	
AIC	50,382.537		21,698.986		28,723.220	

OR: odds ratio; 95%CI: 95% Confidence interval; ICC: intraclass correlation coefficient; AIC: Akaike Information Criterion. * statistically significant results; GMA: Adjusted Morbidity Groups; PCC: complex chronic patient.

## Data Availability

Data are available upon request to the Aragon Health Department.

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
