# Peer review of "Virtual Consultation in Dermatology: Access Inequalities According to Socioeconomic Characteristics and the Place of Residence"

_healthcare, 2024, doi:10.3390/healthcare12060659_

Round 1

Reviewer 1 Report

Comments and Suggestions for Authors

Marco-Ibáñez conducted a cross-sectional study to investigate the associations of sociodemographic and clinical characteristics with dermatology virtual consultation. Overall the manuscript is well-written and this is an interesting study. I have a few comments that I think could help strengthen the presentation of the methods and results.

  • In section 2.3 analysis, please indicate what index was used to determine statistical significance. For example, a 2-sided p-value smaller than 0.05 was used to determine statistical significance.
  • Please describe what parameters were used to summarize categorical and continuous variables respectively. For example, mean and sd were used to describe continuous variables and frequencies and proportions were used to describe categorical variables.
  • Please also describe what bivariate statistical tests you have used to compare categorical and continuous variables between virtual and non-virtual groups.
  • In table 1s, please combine N and % to one column, and change the column name to N(%) / Mean(SD). Similar changes to table 1.
  • Please explain why there are some regions in figure 1a and figure 1b that do not have use rate information. I think it could be more informative if you could also show which regions are rural and which regions are urban areas. I think you could use different borderline colors to reflect that.
  • In lines 177-184, please include p-values at the end of each your statement.
  • please indicate the number of samples included in each of your regression models in table 2.
  • Could you describe how many individuals are included in BDU, and of which how many have data from GMA. Did you have any missing data for sociodemographic and clinical information? What are the reasons that some of the individuals in BDU have missing data in GMA? How did you deal with missing data?

Reviewer 2 Report

Comments and Suggestions for Authors

Thank you for the opportunity to review the article "Virtual consultation in Dermatology: bridging gaps or reinforcing disparities?

The title of the manuscript does not reflect its content. The authors focused on assessing the community's access to virtual dermatology consultations in terms of socioeconomic factors.

My suggestions are to indicate in the paper:

1. what differences in the community influenced the problem of real dermatology consultations;

2. what factors affect the reliability of video dermatology consultations;

3. from what data the authors conclude that socioeconomic status affects the number of video consultations;

4. To what extent and with what devices the consultations took place. Whether the quality of the call influenced the diagnosis;

5. what factors contribute to the number of video consultations and what factors hinder them;

6. The lack of implementation in the work regarding the proposed solutions.

Reviewer 3 Report

Comments and Suggestions for Authors

Thank you for submitting the manuscript of an interesting topic. Please find the following comments:

a) Line 9 of the abstract: correct the thousand separator of 39.174 and 16.910. Both also appeared in section 3 Results.

b) For introduction part, better to include Telemedicine or telehealth as the umbrella term and then explain any differences between teledermatology versus virtual consultation in dermatology

c) It is important for readers from other countries to understand if patients can have the freedom to choose to go virtual or non-virtual or chosen by the GP of the referral. Please add this element into early part of the paper or otherwise the results are meaningless.

d) For section 2.2, add ',' after Regarding inter-consultations, 

e) I don't understand why there are table 2 and table 2 S. Better to simplify all into Table 1, 2, 3 and so on. Same issues applied to Figure.

f) For section 4 about Discussion, In 'last' years should be modified in 'recent' years

Comments on the Quality of English Language

1) please double check the short form of the BDU, GMA, CEICA and PCC as the sequences are not following their full name

2) correct the spanish word of 'Porcentaje' into 'Percentage' in Table 2.S

3) Standardise the use of either 'No Virtual' or 'Non-virtual' among all tables

Round 2

Reviewer 2 Report

Comments and Suggestions for Authors I would like to thank the authors for practical supplementation of the article
with my suggestions. The answers provided by the authors are comprehensive and
factual